# Reduced soil fauna decomposition in a high background radiation area

Hallvard Haanes[1,2]*, Runhild Gjelsvik[1,2]

**1** Norwegian Radiation and Nuclear Safety Authority, Østerås, Norway, **2** Centre for Environmental Radioactivity (CERAD CoE), NMBU, Ås, Norway

* Hallvard.Haanes@dsa.no

## Abstract

Decomposition of litter and organic matter is a very important soil ecosystem function where soil fauna play an important role. Knowledge of the responses in decomposition and soil fauna to different stressors is therefore crucial. However, the extent to which radioactivity may affect soil fauna is not so well known. There are some results showing effects on soil fauna at uranium mines and near Chernobyl from relatively high levels of anthropogenic radionuclides. We hypothesize that naturally occurring radionuclides affect soil fauna and thus litter decomposition, which will covary with radionuclide levels when accounting for important soil parameters. We have therefore used standardised litterbags with two different mesh sizes filled with birch leaves (*Betula pubescens*) to assess litter decomposition in an area with enhanced levels of naturally occurring radionuclides in the thorium ($^{232}$Th) and uranium ($^{238}$U) decay chains while controlling for variation in important soil parameters like pH, organic matter content, moisture and large grain size. We show that decomposition rate is higher in litterbags with large mesh size compared to litterbags with a fine mesh size that excludes soil fauna. We also find that litter dried at room temperature is decomposed at a faster rate than litter dried in oven (60°C). This was surprising given the associated denaturation of proteins and anticipated increased nutritional level but may be explained by the increased stiffness of oven-dried litter. This result is important since different studies often use either oven-dried or room temperature-dried litter. Taking the above into account, we explore statistical models to show large and expected effects of soil parameters but also significant effects on litter decomposition of the naturally occurring radionuclide levels. We use the ERICA tool to estimate total dose rate per coarse litterbag for four different model organisms, and in subsequent different statistical models we identify that the model including the dose rates of a small tube-shape is the best statistical model. In another statistical model including soil parameters and radionuclide distributions, $^{226}$Ra (or uranium precursory radionuclides) explain variation in litter decomposition while $^{228}$Ra (and precursors) do not. This may hint to chemical toxicity effects of uranium. However, when combining this model with the best model, the resulting simplified model is equal to the tube-shape dose-rate model. There is thus a need for more research on how naturally occurring radionuclides affect soil fauna, but the study at hand show the importance of an ecosystem approach and the ecosystem parameter soil decomposition.

**Data Availability Statement:** Raw data has been uploaded to the Mendeley data repository with the following DOI: 10.17632/384dttzrt3.1.

**Funding:** This work was supported by the Research Council of Norway through its Centres of Excellence funding scheme, project number

223268/F50. This was received by HH and RG at Norwegian Radiation and Nuclear Safety Authority (DSA) thorough the Centre for Environmental Radioactivity, CERAD, Centre of Excellence CoE (https://www.nmbu.no/en/services/centers/cerad).

**Competing interests:** The authors have declared that no competing interests exist

## Introduction

Soils provide important ecosystem functions and services like decomposition, nutrient cycling, nitrogen fixation, control of greenhouse gases/$CO_2$ fluxes and food production [1–3]. An important part is their soil fauna, which often is very species rich and links to above-ground biodiversity [4, 5] and land productivity [2, 3]. Soil organisms are important for soil formation and regulate major ecosystem processes like organic matter turnover, nutrient cycling and mineralization, affecting both microbes and plants [6, 7]. Soil fauna affects soil microbes and mineralisation directly by selective feeding, which alters microbe activity, abundance and community, and indirectly through increasing food availability by fragmenting and mixing organic matter in soil [6, 7]. Organic matter is then broken down by microbes which releases nutrients directly to soil or indirectly through symbionts or predators' extrection (nematodes and collembola). This is called mineralization and is of key importance at ecosystem scale because it determines the nutrient availabilty for plant uptake and productivity [6–12].

Knowledge of the responses of soil ecosystems and faunas to different stressors is therefore crucial to mankind to identify any threats to these important ecosystem services. Soil ecosystems are for example often negatively affected by stressors like chemicals through effects on soil fauna activity, abundance and diversity [13–16], but less is known about effects of radionuclides. Assessment of ecosystem responses is however difficult due to their complexity and how functions relate to species richness and interactions among species and food webs [17]. Ecosystem responses can for example not be predicted as the sum of responses in each of the constituent species [18]. Neither ecosystem-level parameters nor indirect effects can be assessed through single species experiments [19, 20]. Examples of important parameters are biodiversity and ecosystem function, which are often linked [21, 22]. In soils such important ecosystem-level parameters includes decomposition and mineralization, which influence land productivity [2].

Litterbag field experiments have shown that soil fauna is important to both decomposition and mineralization rates, depending on climate and litter quality with regard to C:N and lignin content [23–29]. Unambiguous effects have been found in deciduous forest [28], and for birch in different ecosystems and climates [26, 30]. For litter decomposition, meta analyses of litterbag studies show that soil fauna has a significant effect through feeding activity and removal of litter fragments to the soil column where further decomposition occur [7, 27, 28]. Soil fauna generally have a very generalist feeding behaviour and can adapt to abiotic and biotic changes, and there is often functional redundancy with species having overlapping ecological roles [31–33]. Soil fauna can be classified according to their diameter, which is smaller than 100 μm for microflora and microfauna taxa like Bacteria, Fungi, Protozoa and Nematoda, and smaller than 2 mm for mesofauna taxa like Collembola (springtails), Acari (mites), and Enchytraeidae while macrofauna taxa like Lumbricina (earthworms), Isopoda and Mollusca have a wider width [6, 32, 34]. Depending on mesh size, different sized soil fauna can thus enter and affect decomposition and a combination of litterbags with different mesh sizes can be used to assess the decomposing effects of meso and macro soil fauna compared to the effects of microbes and fungus only [25, 29, 35]. It has been show that decomposition in litterbags with different mesh sizes may be affected by differences in microclimate, moisture and aeration relating to mesh size difference that may involve a more efficient microfauna/flora decomposition, except in very dry or wet conditions [23]. Litterbags have also been used to assess effects of anthropogenic radioactive pollution [36, 37].

Negative effects of anthropogenic radioactivity on soil fauna have been extensively reported from spill sites in the old Soviet Union like Chernobyl but with different sensitivities among different soil taxa with earthworms, millipeds, collembolans, enchytraeids and mites being

most sensitive and appropriate as bioindicators [38]. Since these taxa are important to decomposition [23, 39], Chernobyl litterbag results showing reduced litter decomposition with increasing levels of anthropogenic radioactivity levels [36] are not surprising. Ecotoxicological effects on soil fauna have been demonstrated using litterbags and bait lamina in Portugal at uranium contaminated mining sites [40, 41]. However, as far as we know, nobody has attempted to quantify the effects of radioactivity from naturally occurring radioactivity on soil fauna. We hypothesize that naturally occurring radionuclides will affect soil fauna so that litter decomposition is reduced and will covary with radionuclide levels when accounting for important soil parameters. To address this, we used a combination of litterbags with a fine or a coarse mesh to assess litter decomposition along a gradient of naturally occurring radioactivity within an area with elevated background radiation, the Fen igneous complex [42].

## Materials and methods

### Area description

The Fen complex (59.2756˚N, 9.3110˚E) in Norway is the footprint of an eroded volcano (580 Ma) consisting of calcareous carbonatite bedrock types with naturally enhanced radioactivity (Fig 1). One part, Mining hill, consists of red rock with activity concentrations from 670 to 12000 Bq kg$^{-1}$ of $^{232}$Th and from 44 to 550 Bq kg$^{-1}$ of $^{238}$U [43]. Here, bedrock comes to the surface and within Mining hill there are legacy mines and deposits of waste rock. The soil here have much higher than normal levels of the thorium ($^{232}$Th) series radionuclides [44, 45], even though there is large spatial heterogeneity in the distribution of radionuclides [46]. This variation makes Mining hill a suitable place for assessment of soil ecosystem responses through field experiments. It is forested with pine (*Pinus sylvestris*) stands and birch (*Betula pubescens*) is common. Moreover, radionuclides in the soil are known to transfer to earthworms [47]. Also, in Mining hill, outdoor levels of thoron ($^{220}$Rn) and progenies are several orders higher than normal due to both mine ventilation, waste rock deposits [48] and soil exhalation [42], suggesting significant levels in soil gas.

### Litterbags, litter treatment and standardisation

To perform a field experiment within Mining hill, we prepared 100 litterbags. Half of these were made from two layers fine-meshed (0.1 mm) and the other half from coarse-meshed (2 mm) nylon (Elko filtering co) by gluing together the edges of rectangles of one mesh size with Marin & Teknik (Casco; Supplementary), forming an inner size of 15 x 20 cm. The coarse-meshed litterbags were suited with Velcro for easy opening while the fine meshed bags were glued together after filling with litter to ensure no mesofauna penetration.

Litterbag studies have shown that different quality types of litter with different C:N ratios are decomposed differently [35]. We used standardised amounts (d.w) of newly naturally shed leaves from birch (*Betula pubescens*), which is present at each of the experimental localities. This birch species has C:N ratios ranging from 20 to 33 across Europe [49], which is the C:N ratio range that is most affected by soil fauna decomposition effects [28]. Freshly abscised leaves were sampled in an area with normal background radioactivity during litter-fall from a regularly maintained lawn on September 28 the same year as litterbags were deployed. Low levels of radionuclides associated with the sampled leaves were verified for two subsamples by gamma spectrometry (HPGe).

Other studies of litter decomposition have used litter dried at either at room temperature or by higher temperatures in oven. We hypothesize that heat treatment will involve different nutrient availability, which will increase decomposition rate. To assess this, we dried half of the leaves on newspapers at room temperature for five days in an old and very dry storage

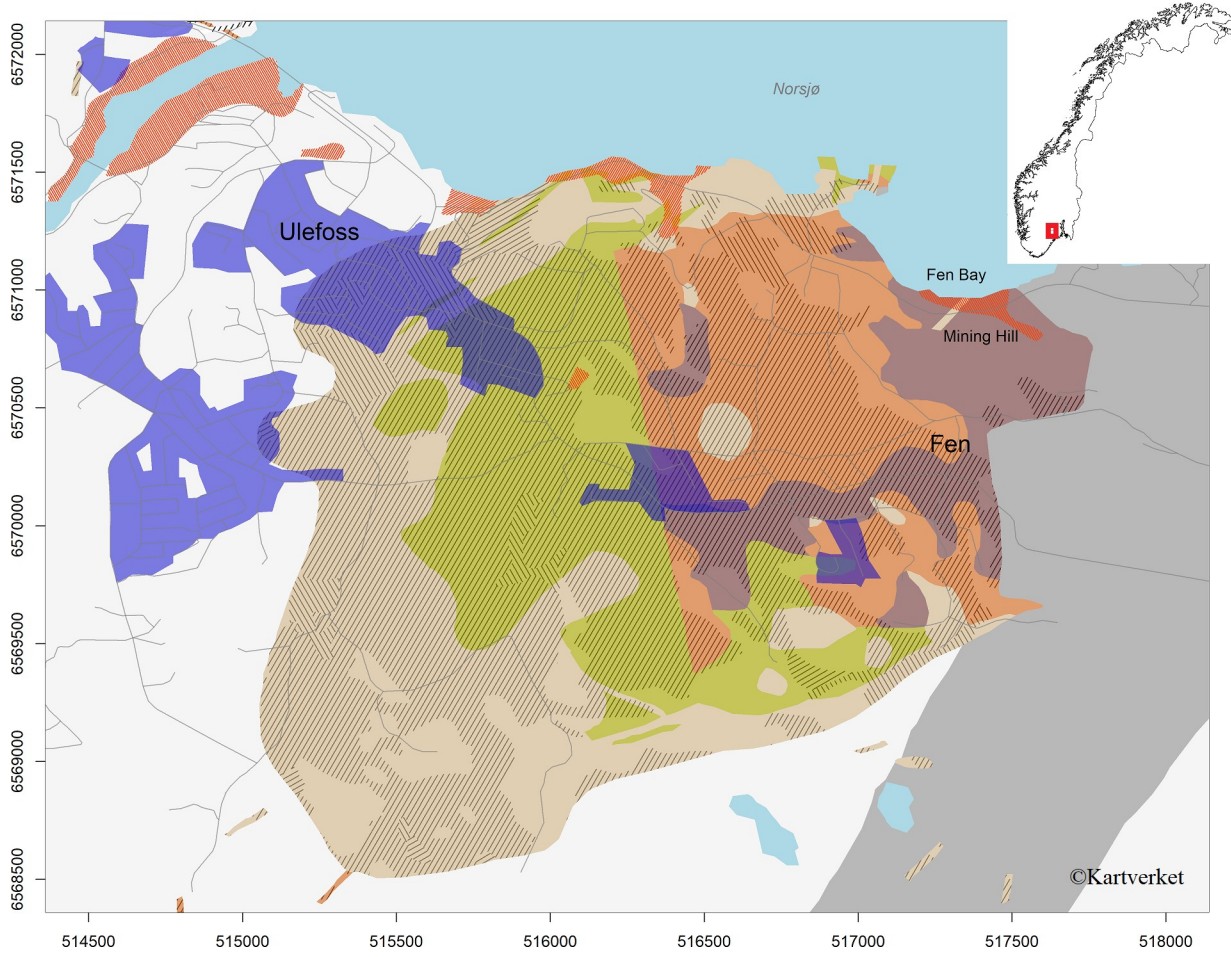

**Fig 1. Map plot of Fen complex.** Hatched markings show Holocene loose mass deposits, orange hatched markings show anthropogenic deposits, blue show densely populated areas and for bedrock rusty is redrock, light chocolate is ankerite, green is søvite, and moccasin is fenite and other Fen bedrock. Generated in R using data from the Geological Survey of Norway (NGU) and the Norwegian Mapping Authority (https://kartkatalog.geonorge.no). X and Y axis are eastern and northern UTM 32 GPS coordinates (WGS84).

facility, while the other half was dried in aluminium trays within paper bags in an oven at 60˚C for 48 hours. After 48 h, the dry weight/wet weight ratio for the oven dried litter (starting wet weights of 200 and 300 grams, mean dry/wet ratio: 0.44, SD: 0.006, n = 2) was equal to the dry weight/wet weight ratio of the room-dried litter (starting wet weights of 55 to 107 grams, mean dry/wet ratio: 0.45, SD: 0.003, n = 3). This is a commonly applied temperature which involves maximum water removal but also denaturation of leaf proteins [50, 51]. For leaves that are heat treated, the extent of protein denaturization increases with temperature as the applied temperature increases from around 30˚C to 80˚C, at which point approximately all protein has denaturized becoming less accessible to extract from the leaf [50, 51]. We wanted to assess whether oven-dried litter is decomposed/lost from litterbags faster than litter dried at room temperature. The two forms of dried litter were therefore applied in the same amount in a balanced experimental design. Afterwards, the fine mesh-sized litterbags were filled with 7.99–8.03 grams of dried litter (mean: 8.00, SD: 0.01), and the coarse-sized litterbags were filled with from 7.77 to 8.00 grams of dried litter (mean:7.86, SD:0.05, Fig 2). The small but significant (t = 22, p<<0.01) difference in litter fill was due to loss of small fragments through the meshes during filling. The litterbags were kept in four large sturdy bags without any holes (IKEA)

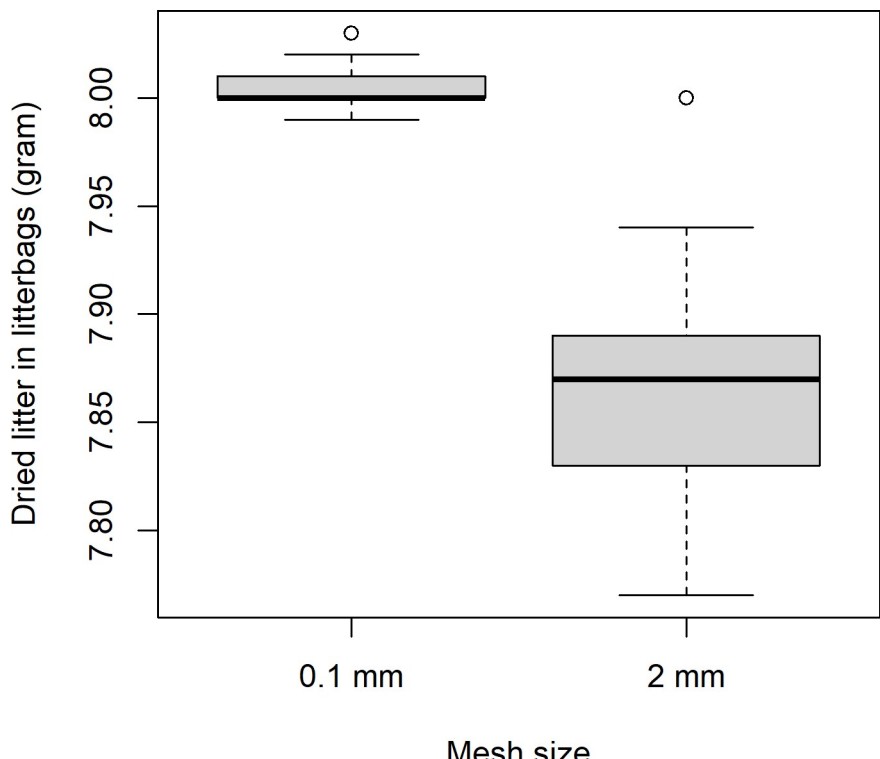

**Fig 2. Boxplot of amount of dried litter filled in fine-meshed and coarser-meshed litterbags before one deployment.**

until being placed on the soil in Fen, and to ascertain that no fragments of litter were lost from the coarse meshed litterbags during transport, the bottom of these bags were carefully inspected after transport.

## Experimental design

Within Mining hill, five localities were established through gamma spectrometry to represent a gradient of soil radionuclides in the $^{232}$Th decay chain. Each locality consisted of a 5x5 meter square within a pine stand of approximately the same height and density. Twenty large litterbags were placed at each locality, two by two into quartets of four litterbags, with five quartets per locality at the centre and corners of the locality square, except at one location (Loc26). At Loc26, one quartet was placed at another location within the square and another quartet was placed right outside the square, and four litterbags in one of the corners were placed four by one (next to each other) due to rocks and very uneven substrate. The pattern of litterbags across localities was situated according to litterbag treatment to achieve a balanced random experimental design. Each quartet consisted of two fine-meshed and two coarse-meshed, each of which contained one litterbag with air-dried and one with oven-dried litter. Fine and coarse litterbags always lay side by side, but elsewise placements within each quartet were random. Within quartets, distances were the same, with diagonal litterbags (centres) separated by 0.4 meters and adjacent ones (centres) separated by 0.2 meters. Within localities, the quartet centres were separated by 1.4 to 5.7 meters (mean: 3.6, SD:1.1). Within the whole study area, the centres of localities were separated by 35 to 340 meters (mean: 180, SD:115). Litterbags positions therefore represents three different spatial scale levels.

All litterbags were deployed exactly a year from October 17 in 2016 to the same date in 2017, which correspond with the period when litterfall ends at this altitude and latitude. All litterbags seemed un-disturbed and were separately put into and transported in aluminium trays. Mesofauna within litterbags was recorded, as remaining litter was carefully removed from each litterbag and dried in oven at 105 ˚C for 24 h and the difference in dry litter weight before and after deployment was used to calculate the amount (grams) of litter loss per litter bag. Since variation among litterbags in initial litter mass was very small and deployment period was equal, the mass loss was used directly in analyses rather than lost fraction of litter per litterbag.

In Norway there is no system for field permits but rather a separate law, the Outdoor recreation act of 1957, which sets forth public right of access, including field work, to outdoors that is not cultivated farmland or pastures (law data: https://lovdata.no/dokument/NL/lov/1957-06-28-16?q=allemannsretten). Moreover, the present study had negligible effects on the ecosystem functioning as it involved only litterbag deployment and soil sample collection on a limited scale.

## Soil sampling and analyses

To be able to account for spatial heterogeneity, soil core (Ø = 5.4 cm with an entrance of 4.8 cm) samples were taken beneath each of the coarse litterbags (0–6 cm) per litterbag quartet. Soil samples varied in wet weight from 36.4 to 138.5 g (mean:86.2, SD:26.9). Each soil sample was dried at 105 ˚C for 24 h, and the mass difference between wet and dry weight of each sample was used to assess its fraction of soil moisture. After drying, samples were sieved (2 mm) and soil particles larger than 2 mm were retained to calculate their fraction of the samples total mass. From the sieved soil (<2 mm), subsamples were taken for assessment of pH, organic matter content and activity concentrations of radionuclides. Soil pH was measured with an inoLab 7110 pH meter according to ISO TC WI: 2003 (CEN/BT/Task Force 151) using a buffer to counter effects of ionic bindings. Levels of $^{232}$Th and $^{238}$U decay series radionuclides in soil samples were assessed through gamma spectrometry using high purity Germanium (HPGe) detectors analysing for $^{228}$Ra (through $^{228}$Ac at the 338, 911 and 969 KeV energies weighted their uncertainties), $^{226}$Ra (through uncertainty-weighted $^{222}$Rn daughters $^{214}$Pb at energies 295 and 352 KeV and $^{214}$Bi at 609 KeV), assuming approximate secular equilibrium for subsequent progeny except $^{210}$Pb and $^{210}$Po. Significant amounts of these latter radionuclides can due to the long half-life of $^{210}$Pb deposit from air and $^{210}$Pb was therefore analysed separately (at energy KeV with a point-source correction for density with regard to the high OM content of some samples), and secular equilibrium was assumed for its progeny $^{210}$Po. Due to a limited number of such detectors, only 20 soil samples were analysed for $^{210}$Pb across localities. In addition, soil samples were analysed for $^{137}$Cs (at energy 661.65 KeV). These HPGe detectors have relative efficiencies of 23% to 50% and cover energies from 20 keV to 3000 keV. They are placed in a low background laboratory and are regularly controlled against a traceable source. Each analysed sample was placed within a circular plastic measurement geometry (36.4 mL), and results corrected for decay since sampling. Within each geometry, the density of each sample was calculated as mass (kg d.w. soil) per volume (36 mL). Finally, organic matter (OM) content was measured by loss on ignition in an oven, using 5 h to reach 550˚C and for 12 h at this temperature, as the difference in mass before and after ignition compared to the dry weight before ignition.

## Dosimetry of soil fauna

The ERICA tool [52–54] was used for dosimetry, modelling organisms in a terrestrial ecosystem with 100% occupancy below ground with the measured soil samples of $^{226}$Ra, $^{228}$Ra and

[137]Cs as input. The measurements of these radionuclides were used as proxy for all the modelled radionuclides, assuming secular equilibrium. In the newest version of the ERICA tool, the contribution of radionuclides in the thorium and uranium decay chains with less than 10 days decay are included in the dosimetry of the previous radionuclide exceeding 10 days half-life [52]. Three radionuclides in the [232]Th decay chain and seven radionuclides in the [238]U decay chain were therefore included in the ERICA modelling, as well as the measured levels of [137]Cs. After ERICA assessment, external and total dose rate was summed up across radionuclides per soil sample.

For this assessment, the default ERICA annelid (5.2 gram, Ksi: 0.1, Chi: 0.1) and detritivore arthropod (0.4 gram, Ksi: 0.35, Chi: 0.18) were modelled, as was also two smaller shapes that better represent the soil mesofauna: a smaller tube (0.4 gram, Ksi: 0.08, Chi: 0.08) and a small box (0.4 gram, Ksi: 0.4, Chi: 0.4). In ERICA an ellipsoid shape is assumed with a given ratio between width and length (Ksi) and a given ratio between height and width (Chi). The mass range allowed in ERICA spans 1.7x10E-4 kg to 6.6 kg, and assuming a 70% water content, this involves minimum dimensions that are 4.5 x 4.5 x 11.3 mm for a rectangular box, or 1.2 x 1.2 x 16 mm for a tube shape. These shapes are similar in shape but larger than a typical Collembola or Enchytraeid. Collembola range from 0.13 to 2 mm in width and 0.16 to 6 mm in length, while Enchytraeids range from 0.5 to 1.3 mm in width and 1 to 40 mm in length and earthworms range from 2 to 20 mm in width and 12 to 80 mm in length [6, 32, 34].

In ERICA tool, the concentration ratio (CR) equals the ratio between activity concentration per wet weight of the organism divided by the activity concentration per dry weight soil. However, there are few data on the CR of small organisms living on or within soil, or their typical water content so that CR could be calculated from transfer factors (TF), which is the ratio between activity concentrations of organisms and soil both per dry weight.

For the annelid, ERICA database CR values were used for all radionuclides except the radium isotopes where the default CR values for similar reference organism are the only choice. For enchytraeids CR's are very scarce and there are no suggestions in ERICA. Among earthworm species, CR's for thorium isotopes vary from 0.006 to 0.024 while uranium CR's range from 0.005 to 0.064 [47, 55, 56]. In ERICA tool, the annelid default CR is 0.009 for thorium isotopes and 0.034 for uranium isotopes. For the smaller tube-shape to represent Enchytraeids, we used literature-based CR's of 0.015 for the thorium isotopes and 0.035 for the uranium isotopes. CR's for lead and radium isotopes are scarce but for one of the earthworm species above, *Eisenia andrei* (50–70 mm length), assuming 80% water content, these values are 0.4 and 0.82, respectively [55]. For the earthworm genus Lumbricina, CR's of 0.095 and 0.096 have been reported for [210]Pb and [210]Po [56]. In ERICA, the default annelid CR's are 0.043, 0.48 and 0.01 for [226]Ra, [210]Pb and [210]Po. For the small tube-shape (Enchytraeid), we used CR's of 0.48 for [210]Pb and 0.01 for [210]Po (similar between ERICA and literature), and 0.08 for [137]Cs (as ERICA annelid CR). However, for the tube-shape we used a literature-based CR of 0.8 for the radium isotopes.

For the detritivore arthropod ERICA database CR values were used for all radionuclides except where missing, as for the polonium isotope where the default CR from a similar reference organism was used and the thorium CR where the default combination method was used. The ERICA detrivore arthropod default CR's are 0.005 for thorium isotopes, 0.01 for uranium isotopes, 0.04 for [210]Pb, 0.01 for [210]Po and 0.11 for [137]Cs. For woodlice and diplopods, literature [226]Ra CR ranges from 0.68 to 1.2 and from 0.54 to 0.79, respectively, when assuming 65% water content [57]. Woodlice have great transpiration power but the water content is typically around 65% for soil arthropods like diplopods and Collembola [56, 58]. For the box shape representing Collembola (and Acari), we therefore used the ERICA CR values for all radionuclides except for the radium isotopes where we used a CR of 0.66 (average diplopod CR).

## Statistical analyses and maps

The fraction of soil fragments >2mm, soil OM, pH and $^{137}$Cs levels, as well as litter loss had an approximate lognormal distribution. Similarly, naturally occurring radionuclides in general have lognormal distributions [59, 60], as have also been shown for the soil $^{232}$Th progeny in the study area [61, 62]. These parameters were therefore $\log_{10}$-transformed (x + 0.001, to avoid log to zero) prior to statistical analyses. It has been highlighted to assess actual weight loss over time and not percent [23]. This especially applies if initial weight or deployment period varies. Since the same litter mass was used to fill each litterbag (but see section 2.2 with regard to a slight loss of litter from coarse mesh bags) and the deployment period was the same for all litterbags, we use lost litter (grams d.w.) rather than rate of loss in our statistical analyses.

To assess spatial autocorrelation, correlations were made between spatial distance and difference in a parameter between soil samples using an array of all possible pairwise physical distances between all soil samples (n = 1225) and correlating it to an array of the corresponding pairwise absolute differences between each of these soil sample pairs for the soil parameter of interest. The distance array was calculated using the dist() function since localities are separated at most by 100's of meters, while the corresponding pairwise parameter differences were calculated with the combn() function, both in base R [63] using Euclidean distances. In addition, range, mean and variation was calculated for the pairwise differences in each soil parameter separately for each spatial scale level, i.e. for litterbag pairs separated by 20–40 cm, pairs separated by 1.4 to 5.7 m, and for pairs separated by 35 to 340 m.

Statistical analyses were done in R [63]. To assess what may explain litter loss, additive linear models of the litter loss ($\log_{10}$-transformed) among litterbags were explored. Explanatory variables included whether litter was dried in oven or room temperature and the soil parameters ($\log_{10}$-transformed), either alone as a null model or together with either activity concentration distributions ($\log_{10}$-transformed) of the radionuclides most important to dose rates, or the dose rates ($\log_{10}$-transformed) estimated at each coarse litterbag through the ERICA tool for either of the four assessed organism types. Each model was simplified removing non-significant terms (potentially explanatory variables), one at a time. Parameter estimates are presented from models after stepwise simplification since this improves accuracy [64]. Models were then compared through Aikakes information criterion (AIC) penalized according to small sample size and number of parameters assessed [65].

A map was made in R using packages SP [66] and RGDAL [67] and map data on bedrock and Holocene loose mass deposition spatial distributions provided by the Geological Survey of Norway (NGU) and basic map data that was all downloaded from the Norwegian Mapping Authority web pages Geonorge (https://kartkatalog.geonorge.no).

## Results

### Soil samples

The soil samples from Mining hill contained fragments larger than 2 mm (sieved off) in a fraction that ranged from less than 0.01 to 0.44 (mean: 0.11, median: 0.08, SD: 0.10), but with some variation among the localities (Fig 3). Soil pH (measured in buffer) ranged from 3.3 to 5.5 (mean: 4.8, median: 4.9, SD: 0.5), with especially one locality standing out (Fig 4). The fraction of organic matter (OM) content (after drying and sieving) ranged from 0.10 to 0.69 (mean: 0.30, median: 0.32, SD: 0.11), also with substantial variation among localities (Fig 5). The soil moisture fraction ranged from 0.29 to 0.55 (mean: 0.38, SD: 0.07), involving similar variation among localities (Fig 6), and had moderate positive correlation to $\log_{10}$-transformed OM (r = 0.63, p<<0.01). Prepared in fixed volumes (36.4 mL) for gamma spectrometry the

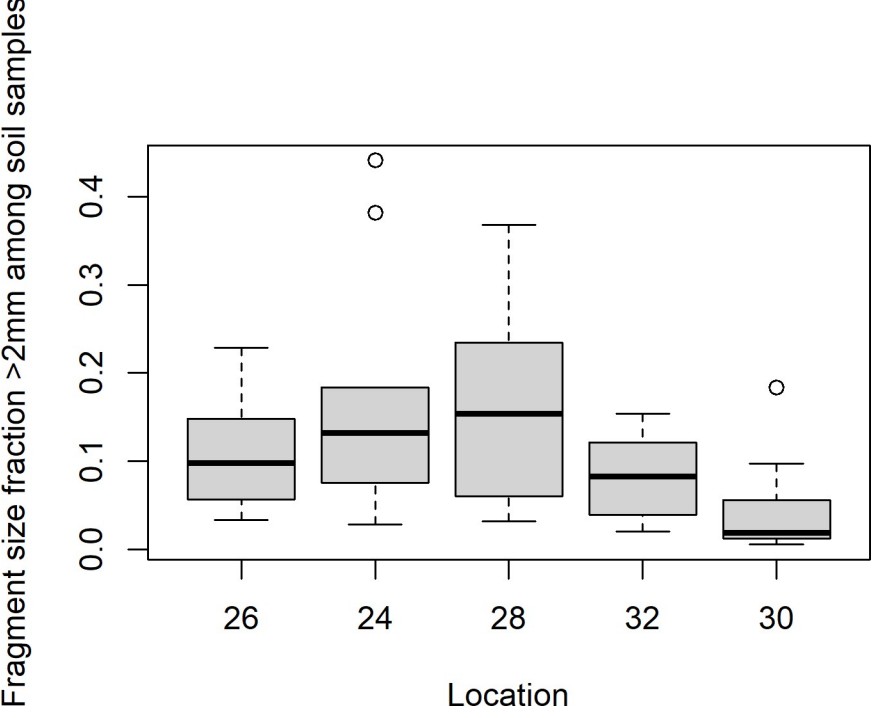

**Fig 3. Boxplot of size fraction >2 mm of soil samples among localities within Mining hill.**

samples weighed from 14 to 34 grams (d.w.), which was strongly negatively correlated to $\log_{10}$-transformed OM (r = -0.96, p<<0.01) and involved densities from 0.39 to 0.95 grams/mL (mean: 0.64, SD: 0.16). The content of $^{228}$Ra ranged from 470 to 5200 Bq kg$^{-1}$ soil dry weight (mean: 2400, median: 2700, SD: 1400), with much variation and little overlap between localities (Fig 7). Levels of $^{226}$Ra ranged from 24 to 120 Bq kg$^{-1}$ soil dry weight (mean: 45, median:

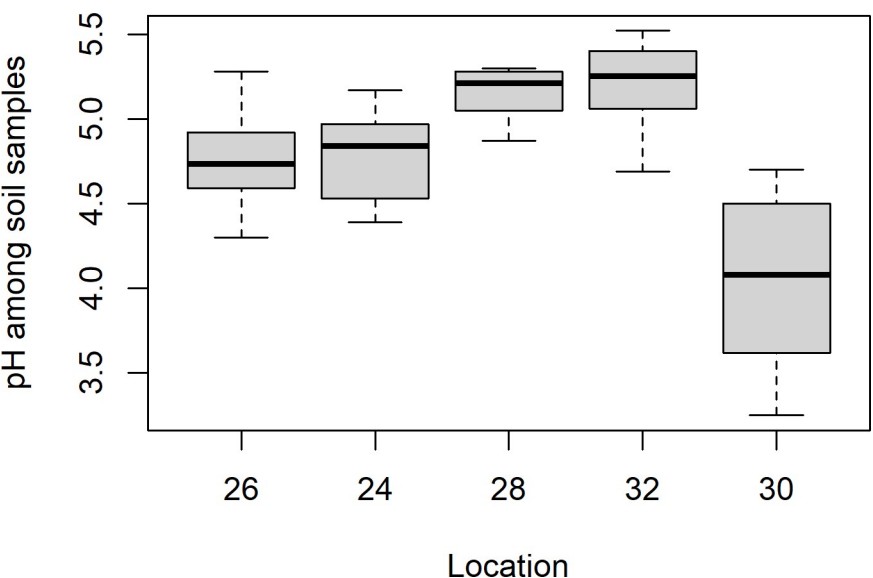

**Fig 4. Boxplot of pH of soil samples among localities within Mining hill.**

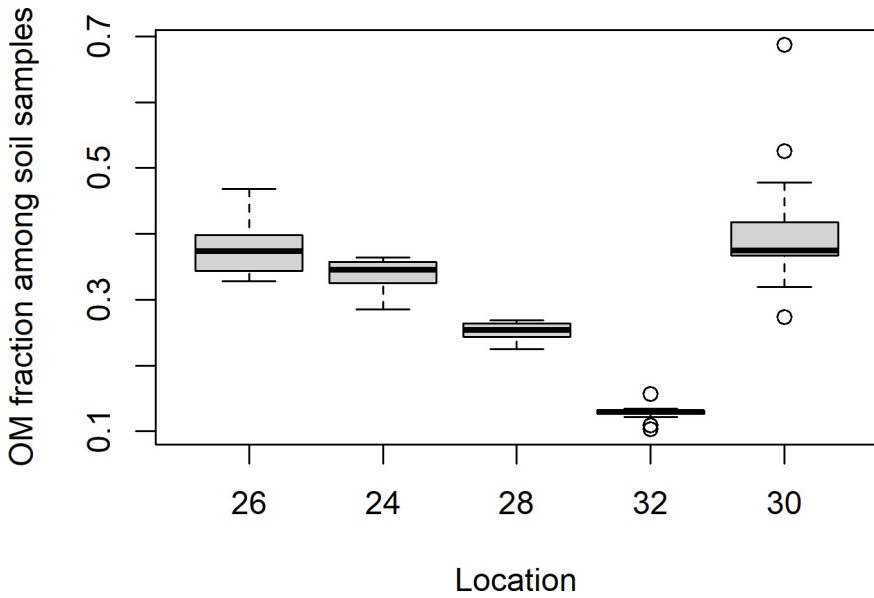

**Fig 5. Boxplot of organic matter (OM) fraction of soil samples among localities from Mining hill.**

32, SD: 24), with some variation among localities and especially much within locations 26, 30 and 32 (Fig 8). Assuming secular equilibrium, the ratio between these radium isotopes in the soil samples show that within the study area the activity concentration is from 10 to 141 (mean: 58) times higher for the [232]Th decay series radionuclides than for the [238]U series radio-nuclides. Levels of [210]Pb (n = 20) were for all except one locality higher than [226]Ra levels (Fig 9), and for samples with no analysed [210]Pb, the location-wise ratio of [210]Pb to [226]Ra was used to estimate [210]Pb from the analysed [226]Ra value.

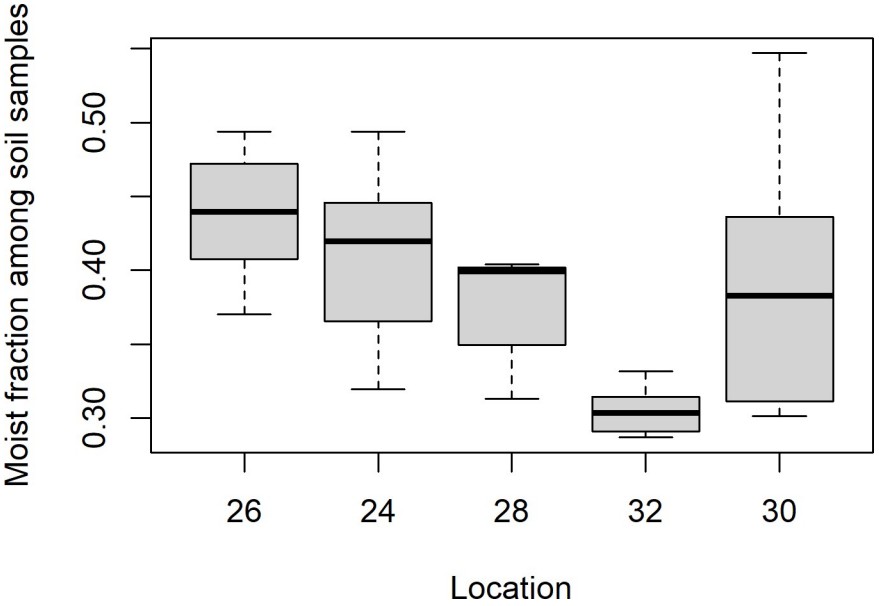

**Fig 6. Boxplot of moist fraction of soil samples among localities within Mining hill.**

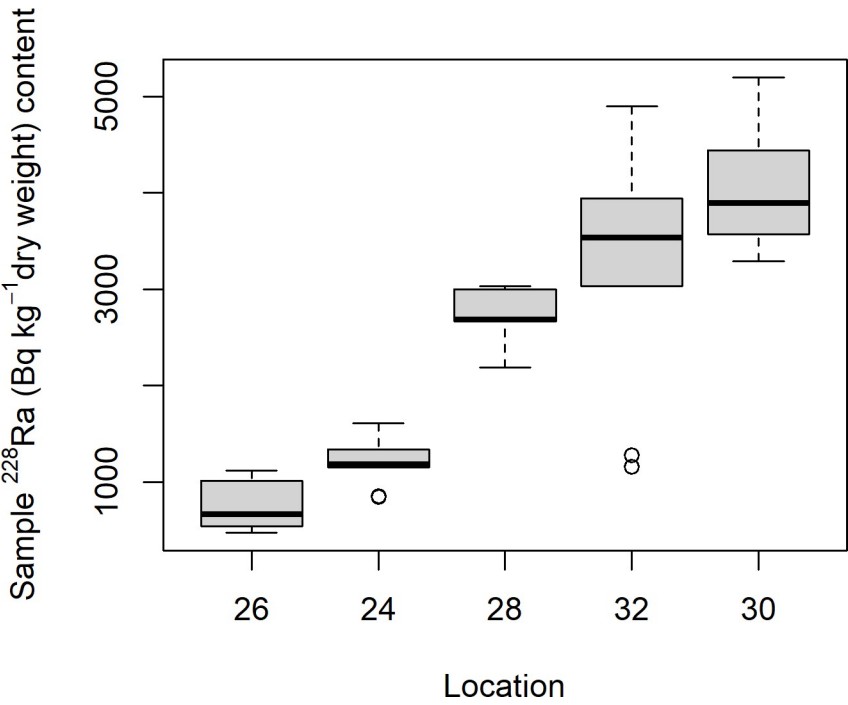

**Fig 7. Boxplot of $^{228}$Ra (Bq kg$^{-1}$ d.w.) of soil samples among localities within Mining hill.**

The log$_{10}$-transformed $^{228}$Ra levels were weakly correlated to log$_{10}$-transformed levels of $^{226}$Ra (r = 0.41, p<0.01), negatively weakly correlated to log$_{10}$-transformed OM (r = -0.36, p<0.02) but not to log$_{10}$-transformed soil pH (r = -0.13, p>0.4). The log$_{10}$-transformed $^{226}$Ra

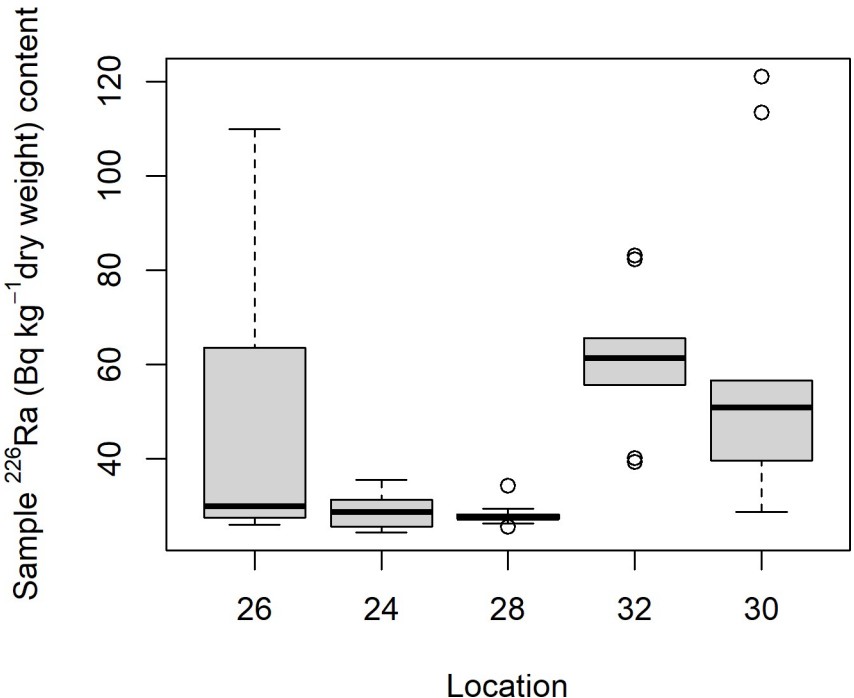

**Fig 8. Boxplot of $^{226}$Ra (Bq kg$^{-1}$ d.w.) of soil samples among localities within Mining hill.**

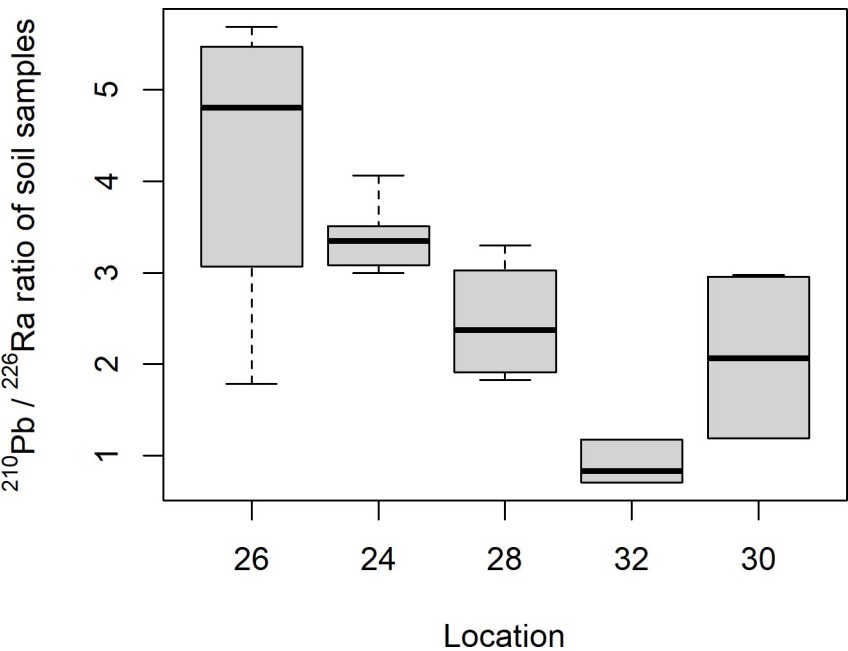

**Fig 9. Boxplot of $^{210}$Pb to $^{226}$Ra ratio of soil samples among localities within Mining hill.**

was also weakly negatively correlated to $\log_{10}$-transformed OM (r = -0.29, p<0.05) but neither to $\log_{10}$-transformed soil pH (r = 0.09, p>0.5). $\log_{10}$-transformed soil particle fraction >2mm was negatively weakly correlated to $\log_{10}$-transformed $^{226}$Ra (r = -0.31, p<0.05) and almost to $^{228}$Ra (r = -0.26, p = 0.07). Soil levels of $^{137}$Cs ranged from 15 to 120 Bq kg$^{-1}$ soil dry weight (mean: 35, SD: 19), also with considerable variation among localities but at very low levels (Fig 10).

Regarding spatial autocorrelation, all soil parameters had significant but at varying strengths correlations to the physical distances between pairs of soil samples, while litter loss itself did not (Table 1). This included a weak negative spatial autocorrelation for $\log_{10}$ of soil pH and a weak positive spatial autocorrelation for $\log_{10}$ of both soil $^{228}$Ra and soil $^{226}$Ra, while no spatial autocorrelation at all for litter loss (only coarse litterbags). The pattern is also clear among the three levels of spatial scales, as the means and standard deviations of soil $^{228}$Ra pairwise absolute differences increase by doubling and tripling, respectively, with spatial scale (from within quartets, to within locations and within study area, Table 2). By comparison, the mean and standard deviation of pairwise absolute difference of soil $^{226}$Ra also increased from within quartets to within localities but not from within localities to within the study area, expressing spatial autocorrelation but a less clear gradient among localities. The $^{137}$Cs absolute differences between soil samples have similar means and standard deviations between the three scale levels (Table 2).

## Dosimetry

External dose rates estimated for the default annelid with ERICA ranged from 0.7 to 6.8 μGy h$^{-1}$ (mean: 3.1, SD: 1.8) with clear differences between the localities (Fig 11), and were very similar in the other three modelled organisms, differing maximally by 0.016 μGy h$^{-1}$ for any radionuclide or on average only 0.001 μGy h$^{-1}$ across radionuclides. Including internal dose

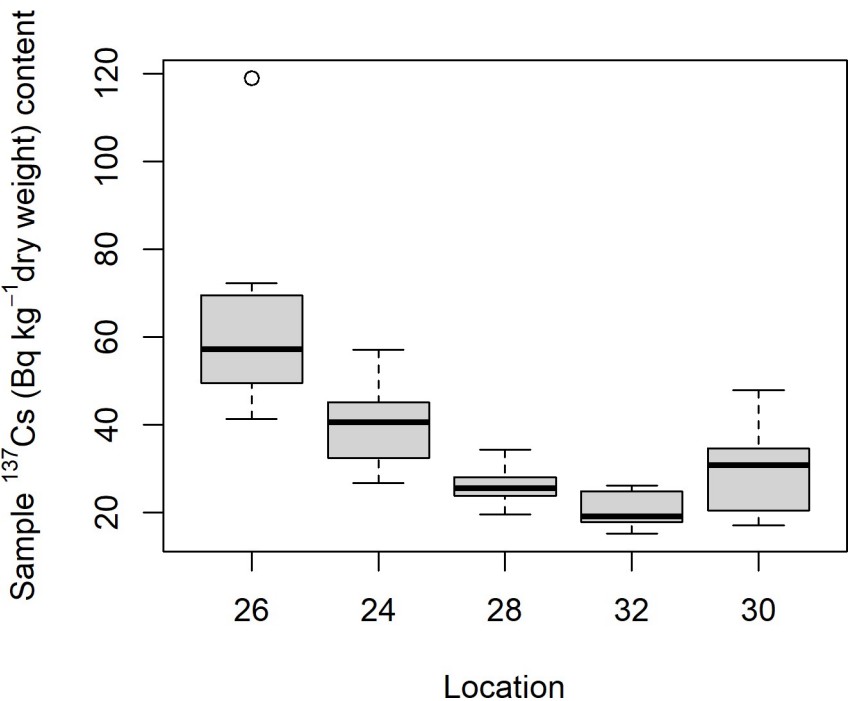

**Fig 10. Boxplot of $^{137}$Cs (Bq kg$^{-1}$ d.w.) of soil samples among localities within Mining hill.**

rates, the total dose rates varied between the four modelled organisms and ranged from 1.8 to 17 µGy h$^{-1}$ (mean: 8.2, SD: 4.6) for the annelid, from 1.4 to 13 µGy h$^{-1}$ (mean: 6.1, SD: 3.4) for the detrivore arthropod, from 5.4 to 32 µGy h$^{-1}$ (mean: 16, SD: 8.0) for the small tube shape and from 3.8 to 20 µGy h$^{-1}$ (mean: 10, SD: 4.9) for the box shape. The pattern across localities was relatively similar for the four organisms (Fig 12) but with higher levels and more variation in the two additional shapes. Among radionuclides, the main contributions to dose rate came from the radionuclides in the $^{232}$Th chain and from $^{226}$Ra (Table 3). Differences between the four organisms in the radium isotopes are due to the different applied CR's. For the small tube and box shapes, the higher radium isotope CRs result in higher dose rates but notice the levels of these dose rates for $^{226}$Ra and $^{228}$Ra compared to the levels of activity concentrations (being due to the large difference in half-life).

**Table 1. Correlation strength and significance.**

| Correlation | Strength (r) | Significance (p) |
|---|---|---|
| *Distance (m) * Δ fraction fragments >2mm* | -0.18 | <<0.01 |
| *Distance (m) * Δ soil moisture (fraction)* | -0.09 | <<0.01 |
| *Distance (m) * Δ log$_{10}$ soil OM (fraction)* | 0.08 | <<0.01 |
| *Distance (m) * Δ log$_{10}$ soil pH* | -0.34 | <<0.01 |
| *Distance (m) * Δ log$_{10}$ soil $^{228}$Ra (Bq kg$^{-1}$ d.w.)* | 0.37 | <<0.01 |
| *Distance (m) * Δ log$_{10}$ soil $^{226}$Ra (Bq kg$^{-1}$ d.w.)* | 0.31 | p<<0.01 |
| *Distance (m) * Δ log$_{10}$ soil $^{137}$Cs (Bq kg$^{-1}$ d.w.)* | -0.15 | <<0.01 |
| *Distance (m) * Δ litter loss (gram year$^{-1}$)* | 0.03 | >0.29 |

Between physical distance of all pairs of soil samples and corresponding difference (Δ) of each parameter.

**Table 2. Spatial heterogeneity.**

|  | Within quartets | | | Within localities | | | Within study area | | |
|---|---|---|---|---|---|---|---|---|---|
|  | Range | Mean | SD | Range | Mean | SD | Range | Mean | SD |
| $\Delta$ Soil $^{228}$Ra (Bq kg$^{-1}$ d.w.) | 0–1210 | 250 | 320 | 0–3700 | 610 | 720 | 0–4700 | 1600 | 1200 |
| $\Delta$ Soil $^{226}$Ra (Bq kg$^{-1}$ d.w.) | 0.8–35 | 5 | 7 | 0–92 | 18 | 23 | 0–97 | 24 | 24 |
| $\Delta$ Soil $^{137}$Cs (Bq kg$^{-1}$ d.w.) | 0.1–71 | 9 | 14 | 0–78 | 11 | 13 | 0–103 | 19 | 19 |

For $^{228}$Ra, $^{226}$Ra and $^{137}$Cs: range, mean and standard deviation (SD) of absolute differences between all pairs of soil samples within each quartet (smallest spatial scale), within each locality (medium spatial scale), as well as within the whole study area (largest spatial scale).

## Litter loss

After recollection, no mesofauna was found within fine meshed litterbags but in four of the coarse meshed litterbags (totally 13 enchytraeids and three small earthworms). Within Mining hill, large litter bags had lost from 2.5 to 6.1 grams (d.w.) litter (mean: 3.6, SD: 0.7) after one year of deployment during the field experiment. For the coarse-meshed litterbags this involved a fractional loss (d.w.) ranging from 0.32 to 0.78 (median: 0.49, mean: 0.51, SD: 0.10) while for the fine-meshed litterbags the corresponding lost fraction ranged from 0.33 to 0.56 (median: 0.38, mean: 0.39, SD: 0.04). There was a clear difference in litter lost between coarse and fine meshed litterbags (t = 8, p<<0.01), as well as between litterbags with litter dried in oven and room temperature (t = 3, p<0.01, Fig 13). The latter difference was found among both coarse and fine meshed litterbags. The coarse meshed litterbags lost from 2.5 to 6.1 grams of litter (d. w., loss mean: 4.0, SD: 0.8), while the fine meshed litterbags lost from 2.6 to 4.5 grams litter (d. w., loss mean: 3.1, SD: 0.4). Within quartets, pairs of coarse meshed litterbags with air-dried and oven-dried litter differed with 0.1 to 2.5 grams in loss of litter (d.w., loss mean: 0.7, SD: 0.6), while the corresponding fine meshed litterbag pairs with air-dried and oven-dried litter within quartets differed with 0.1 to 1.3 grams in loss of litter (d.w., loss mean: 0.4, SD: 0.3). By

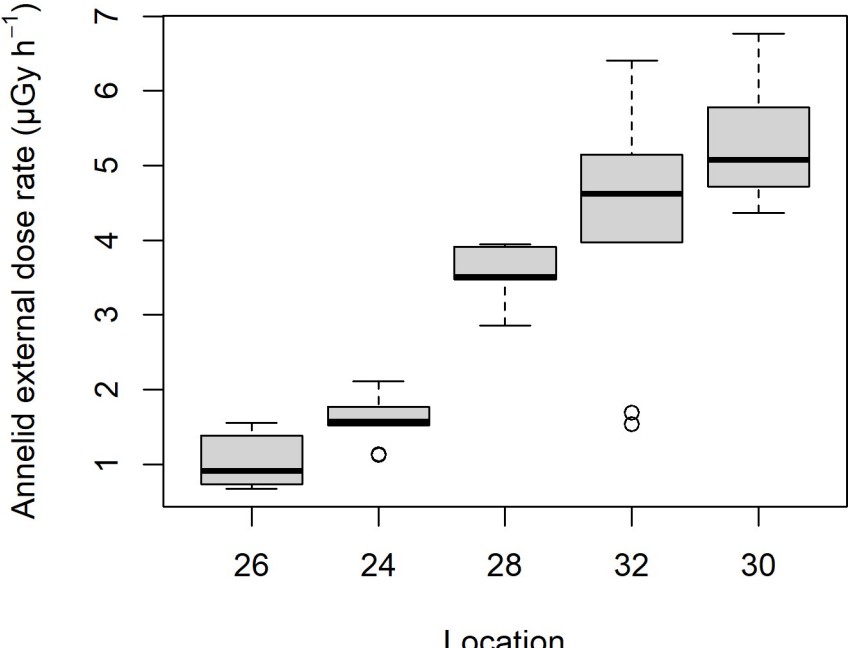

**Fig 11. Boxplot of external dose rate estimated with the ERICA tool for an annelid.**

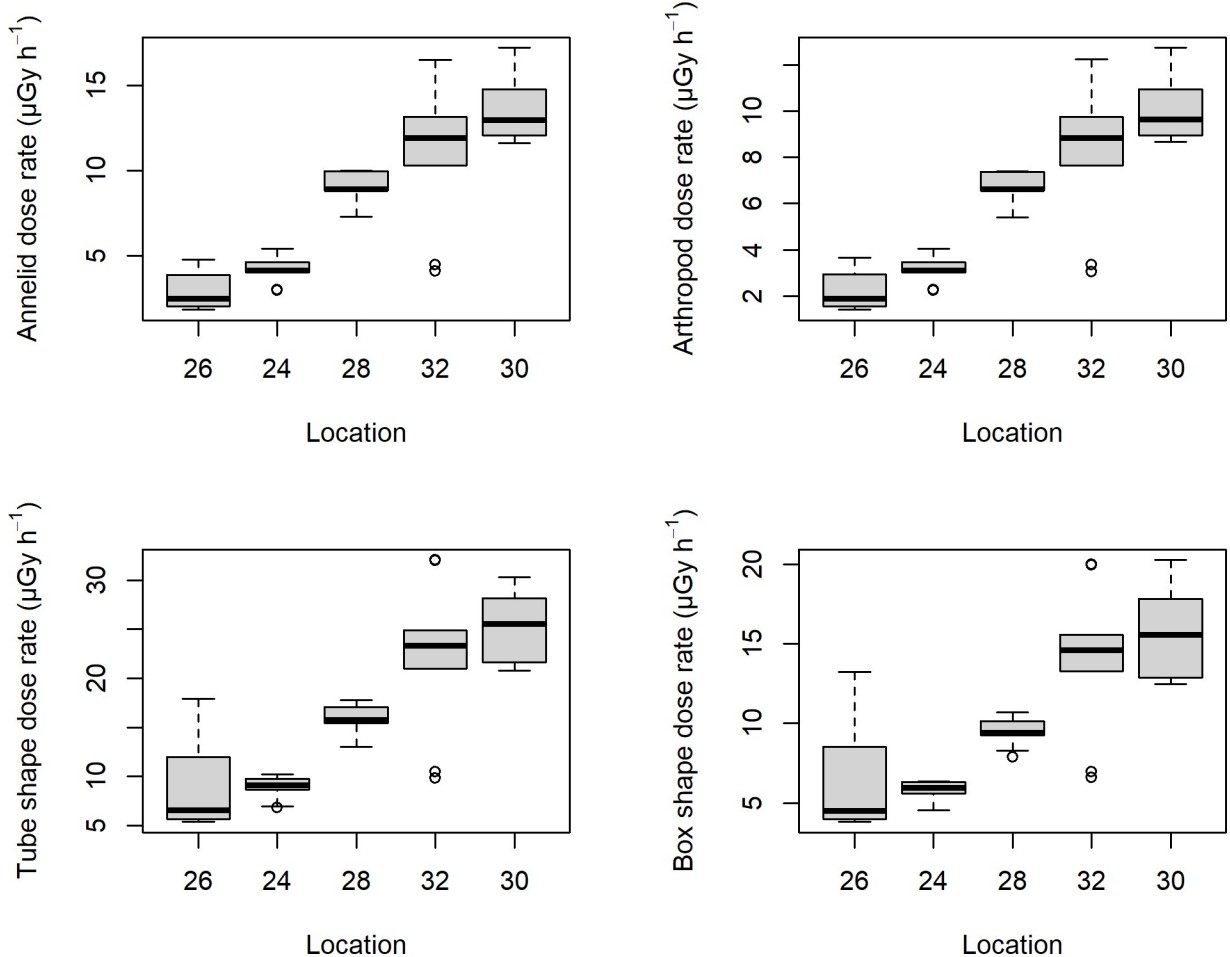

**Fig 12. Boxplots of the total dose rate among localities for the four organisms modelled in ERICA tool.**

comparison, within quartets, pairs of litterbags with oven-dried litter with either coarse or fine mesh differed with from 0.1 to 1.7 grams litter loss (d.w., loss mean: 0.8, SD: 0.4) while pairs of litterbags with air-dried litter with either coarse or fine mesh differed with from 0.1 to 2.6 grams of litter loss (d.w., loss mean: 1.1, SD: 0.7). Differences between litterbag litter loss was thus significant within quartets (relating to mesh size and drying regime) but larger among quartets and localities.

For the linear statistical model of coarse meshed litterbag loss including radionuclide distributions, the radium isotopes were chosen since these contribute most to dose rates and covary with those other radionuclides that also contribute significantly (Table 3). This model had several non-significant terms (Table 4) and after simplification only included $^{226}$Ra and not $^{228}$Ra. Since much dose contribution is from the $^{232}$Th chain (Table 3), an alternative model including only $^{228}$Ra was therefore also subsequently explored through AIC. For the four linear statistical models of litter loss including the estimated dose rates of organisms types (adj $R^2$ = 0.33–0.40, F(36, 7) = 4.5–6.0, p<0.002), non-significant terms included moist (t = 1.2–1.3, p>0.21) for all models while size particle fraction > 2mm ($\log_{10}$-transformed) was non-significant (t = -1.6, p>0.11) for the two default organism models, nearly significant for the small tube-shape (t = -2.0, p<0.052) and significant for the box shape (t = -2.2, p<0.04). After model simplification, the six explored models were far better than the null model but their ability to explain

**Table 3. Radionuclide dose rates.**

|  | Annelid | Arthropod | Small tube | Small box |
|---|---|---|---|---|
| *Th-232* | 0.10–1.1 (0.51) | 0.06–0.61 (0.28) | 0.16–1.8 (0.83) | 0.05–0.60 (0.28) |
| *Ra-228* | 0.24–2.7 (1.2) | 0.24–2.7 (1.2) | 0.32–3.6 (1.6) | 0.31–3.4 (1.6) |
| *Th-228* | 1.2–13 (6.0) | 0.82–9.0 (4.2) | 1.7–19 (8.6) | 0.81–9.0 (4.1) |
| $^{232}$*Th subtotal* | **1.5–17 (7.7)** | **1.1–12 (5.7)** | **2.2–24 (11)** | **1.2–13 (6.0)** |
| *U-238* | 0.02–0.10 (0.04) | 0.01–0.03 (0.01) | 0.02–0.10 (0.04) | 0.01–0.03 (0.01) |
| *Th-234* | 0 | 0 | 0 | 0 |
| *U-234* | 0.02–0.11 (0.04) | 0.01–0.04 (0.01) | 0.02–0.12 (0.04) | 0.01–0.03 (0.01) |
| *Th-230* | 0.01–0.03 (0.01) | 0–0.02 (0.01) | 0.01–0.05 (0.02) | 0–0.02 (0.01) |
| *Ra-226* | 0.17–0.82 (0.31) | 0.17–0.82 (0.31) | 2.7–14 (5.0) | 2.3–11 (4.2) |
| *Pb-210* | 0–0.05 (0.01) | 0–0.04 (0.01) | 0–0.04 (0.01) | 0–0.04 (0.01) |
| *Po-210* | 0.01–0.15 (0.03) | 0.01–0.15 (0.03) | 0.01–0.14 (0.03) | 0.01–0.14 (0.03) |
| $^{238}$*U Subtotal* | **0.28–1.3 (0.44)** | **0.18–1.1 (0.37)** | **2.8–14 (5.1)** | **2.3–11 (4.3)** |
| *Cs-137* | 0–0.04 (0.01) | 0–0.04 (0.01) | 0–0.04 (0.01) | 0–0.04 (0.01) |

Range (mean) of dose rates in µGy h$^{-1}$ (external and internal combined) per radionuclide estimated with ERICA tool for the default annelid and detrivore arthropod, as well as for a small tube and box shape representing an Enchytraeid and a woodlouse living within the Fen soil levels.

variation in loss from litterbags (log$_{10}$-transformed) is relatively similar with the two best being the one including the dose rate of the tubes-shaped organism and the model including the $^{226}$Ra distribution (Table 5). For all six models, the effect of all significant parameters was negative and relatively similar. Among parameter estimates the effect is clearly larger for the soil parameters organic matter fraction (OM) and pH (both log$_{10}$-transformed) but somewhat less for log$_{10}$$^{226}$Ra and especially the dose rate (Table 6). By comparison, the parameter estimate of log$_{10}$$^{228}$Ra was even lower (β = -0.07 SE = 0.04). Combining the two best models into

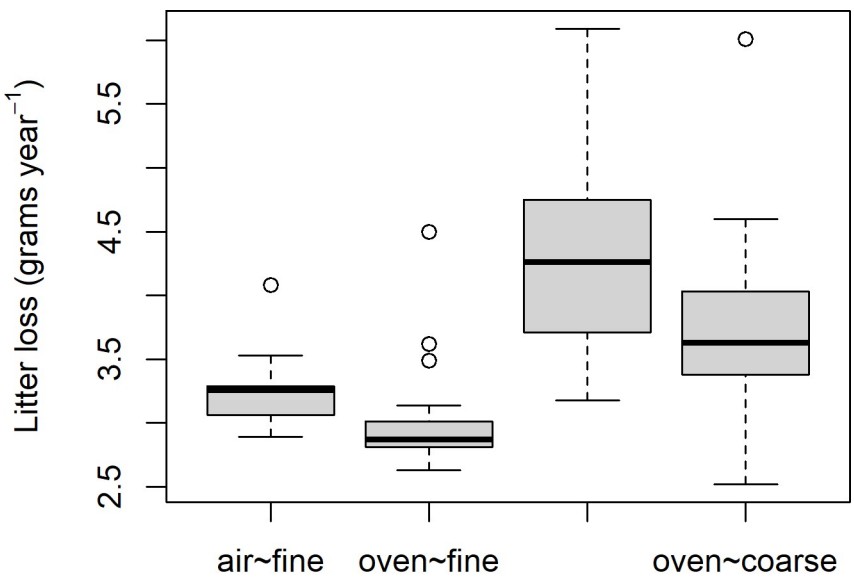

**Fig 13. Boxplot of litter loss.** During one year of exposure from litterbags with coarse (2 mm) and fine (0.1 mm) mesh size when litter was either dried in oven (60˚C) or at room temperature (20 ˚C).

**Table 4. Full versions of the two best statistical models.**

| | AC MODEL | | | | DR MODEL | | | |
|---|---|---|---|---|---|---|---|---|
| | **P** | **SE** | **t** | **p** | **P** | **SE** | **t** | **p** |
| **INTERCEPT** | 1.07 | 0.33 | 3.2 | <0.003* | 0.79 | 0.22 | 3.6 | <0.001* |
| **DRIED_OVEN** | -0.07 | 0.02 | -3.5 | <0.002* | -0.07 | 0.02 | -3-5 | <0.002* |
| **$H_2O$** | 0.26 | 0.21 | 1.2 | <0.24 | 0.27 | 0.21 | 1.2 | <0.23 |
| **OM** | -0.35 | 0.12 | -3.1 | <0.05* | -0.39 | 0.09 | -4.3 | <0.001* |
| **PH** | -0.38 | 0.28 | -1.4 | <0.19 | -0.55 | 0.26 | -2.1 | <0.045* |
| **TUBESHAPE DR** | | | | | -0.07 | 0.02 | -3.0 | <0.005* |
| **$^{228}$RA** | -0.07 | 0.05 | -1.4 | <0.17 | | | | |
| **$^{226}$RA** | -0.15 | 0.07 | -2.3 | <0.03* | | | | |
| **$^{137}$CS** | -0.06 | 0.09 | -0.6 | <0.56 | | | | |
| **>2MM** | -0.07 | 0.03 | -2.2 | <0.04* | -0.06 | 0.03 | -2.0 | <0.055 |

Parameters estimate (P), standard error (SE), t value (t), probability (p) and significance (*) for the models on activity concentration (AC model: adj $R^2$ = 0.39, $F(35, 6)$ = 4.4, p<0.001) and tube shape organism dose rate (DR model: adj $R^2$ = 0.39, $F(37, 6)$ = 5.6, p<0.001) for variation in $\log_{10}$ litter loss, with $\log_{10}$ transformation of fraction of organic matter (OM), pH, fraction of soil particle size>2mm (>2mm), activity concentration of $^{226}$Ra, $^{228}$Ra and $^{137}$Cs (AC mod), total dose rate from all radionuclides (DR mod), but no transformation for litter drying regime (dried_oven) and soil moisture ($H_2O$).

one model yielded the non-significant terms $\log_{10}{}^{226}$Ra (t = -1.6.0, p<0.128), $\log_{10}$pH (t = -1.6.0, p<0.122), $\log_{10}$tubeshapeDR (t = -1.7.0, p<0.103) and $\log_{10}$>2mm (t = -2.0, p<0.056), and ended after model simplification again with the best dose rate model (tubeshape DR).

## Discussion

As expected, there was a much larger loss of litter from litterbags with a coarse mesh size (2 mm) than from litterbags with a fine mesh size (0.1 mm, Fig 2). Presence of macrofauna like earthworms has a substantial effect on decomposition rate, either directly through litter feeding and/or through indirect effects on micro and mesofauna [39, 68], and in a mixed forest ecosystems, earthworms can comsume the entire annual litterfall [6, 69]. An effect of the exclusion of soilfauna access to the litter in the fine-meshed litterbags is therefore anticipated. In addition, differences in mesh size may be attributable to differences in microclimate, moisture and aeration that may affect microfauna and microflora decomposition differently, except in very dry conditions [23]. It is therefore important to assess different mesh sizes and compensate for spatial variation by a high enough spatial resolution when deploying litterbags [27].

**Table 5. Comparison of statistical null models (Null), activity concentration models (AC) and dose rate models (DR).**

| Model | Model terms | Adj $R^2$ | F (df, df) | p-value | AIC |
|---|---|---|---|---|---|
| DR | Dried + OM + pH + tube-shape DR | 0.36 | 7.4 (4, 41) | <0.001 | -112.8 |
| AC | Dried + OM + $^{226}$Ra + >2mm | 0.38 | 7.9 (4, 40) | <0.001 | -110.5 |
| DR | Dried + OM + pH + arthropod DR | 0.32 | 6.3 (4, 41) | <0.001 | -109.8 |
| DR | Dried + OM + pH + annelid DR | 0.32 | 6.3 (4, 41) | <0.001 | -109.6 |
| AC | Dried + OM + $^{228}$Ra + >2mm | 0.31 | 6.0 (4, 41) | <0.001 | -108.8 |
| DR | Dried + OM + pH + box-shape DR + >2mm | 0.40 | 6.9 (5, 38) | <0.001 | -107.1 |
| Null | Dried + OM + pH + + >2mm + $H_2O$ | 0.26 | 4.1 (5, 38) | <0.005 | -98 |

Explored simplified linear models ranked according to penalised AIC. $\log_{10}$ transformation was done for fraction of organic matter (OM), pH, fraction of soil particle size>2mm (>2mm), $^{228}$Ra, $^{226}$Ra, and for the dose rate estimates of each organism type (annelid DR, arthropod DR, tube-shape DR, box-shape DR), not for factorial litter drying regime (dried).

**Table 6. Parameter estimates (β±SE) with the two best simplified linear models of $\log_{10}$ litter loss, with $\log_{10}$ transformation of fraction of organic matter (OM), pH, fraction of soil particle size>2mm (>2mm), $^{226}$Ra and dose rate of organism (tube-shape DR), but not for litter drying regime (dried_oven).**

| Parameter name | Tube-shape DR model, AIC = -113 | $^{226}$Ra model, AIC = -110 |
|---|---|---|
| Intercept | 1.04 ± 0.17 | 0.72 ± 0.08 |
| Dried_oven | -0.06 ± 0.02 | -0.07 ± 0.02 |
| OM | -0.32 ± 0.07 | -0.24 ± 0.06 |
| pH | -0.64 ± 0.24 | - |
| Tubeshape DR | -0.06 ± 0.02 | - |
| $^{226}$Ra | - | -0.19 ± 0.06 |
| >2mm | - | -0.06 ± 0.02 |

However, the direction of the clear effect of the drying regime of the litter among litterbags with both fine and coarse mesh size (Fig 13) and across statistical models was somewhat surprising (Table 6). It is widely known across ecosystems that litter decomposition varies among litter species depending on and increasing with litter quality and levels of C, N and lignin [68, 70–72]. Since drying in oven at 60 ˚C involves denaturation of one third of leaf proteins [50], we anticipated an effect relating to an increased decomposition due to nutrients being more available. The opposite direction of the effect may be due to the increased stiffness of litter after oven drying, possibly making the litter less available to soil fauna fragmentation but could also be due to denaturized proteins being less available to soil fauna digestion. The difference between room-dried and oven-dried litter also among the fine meshed litterbags could support either of these explanations, either that brittleness decreases surface area or that denaturization also make proteins less available to microbes. However, the difference in decomposition rate is of interest in itself since some litterbag studies use only oven dried litter while other studies use room-dried litter, and knowledge about the size of this effect is important when such studies are compared.

The most important soil parameters were assessed to avoid confounding factors and showed like radionuclide distributions different variation and gradients across locations and litterbags (Figs 3–10), involving an ideal opportunity to separate any effects in a statistical analysis. Within the area bedrock consist of redrock, which is a carbonatite, probably involving less variation in soil parameters than if more bedrock were present. This field experiment can thus with its limited spatial extent be viewed as a common garden experiment, where the design with systematised different distances between litterbags allowed assessment and verification of spatial autocorrelation in the most important parameters. The carbonatite bedrock also explains the relatively high soil pH. Also, the negative effect of soil pH on soil fauna is well known [73]. The fraction of organic matter (OM) observed in the study area is relatively high and probably does not per se involve a negative effect on soil fauna. The negative relationship between litter loss and OM in statistical models demonstrate a higher level of OM beneath litterbags with the lowest degrees of decomposition. This is probably due to a lower abundance or activity of soil fauna, as organic matter will build up when decomposition is low. Finally, and most tantalising, when taking into the account the covariation of important soil parameters, there was a clear statistical effect of the radionuclide levels in this area.

However, the results of the statistical model for radionuclide distribution were surprising given their activity concentrations in the study area and the associated dose rates (Table 3). It was surprising that $^{226}$Ra was a highly significant term while $^{228}$Ra was not. This strongly hints to other effects on litter decomposition than just dose rates. The sizes of the effect of terms on litter loss also support this notion, with the $^{226}$Ra term having an effect around five times larger than both the dose rate terms and the $^{228}$Ra term in the statistical models. However, when the

two best models were combined and model simplification performed, the subsequent model was the same as the best dose rate model, highlighting an effect of radionuclide dose rate on soil fauna in the area. Interestingly, removal of the [226]Ra term resulted in the pH term becoming significant, suggesting these terms explaining similar parts of variation in litter loss. One potential explanation is that the concentration ratios (CR) for radium applied in the ERICA tool (see methods section) were too low. The much longer half-life of [226]Ra compared to [228]Ra involves much higher impact on dose rates if higher CR values are used (see Table 3). Thus, if CR values for radium in real are higher than assumed (see methods section), this could explain the higher explanatory power of [226]Ra (and the [238]U series radionuclides) compared to [228]Ra (and the [232]Th series) with much higher activity concentrations.

Another potential explanation for the significance of the term on activity concentration of [226]Ra but not [228]Ra, may be that radionuclide toxicity also is important to soil fauna. Within the study area the activity concentrations of [228]Ra are from 10 to 141 times greater than for [226]Ra, while the half-life of [226]Ra (1600 y) is 280 times longer than for [228]Ra (5.7 y). By comparison, progenitor radionuclides of [226]Ra, uranium isotopes [234]U and [238]U, have half-lives of 250 ky and 4.5E9 years. The number of radionuclide atoms present and thus toxicity of these radionuclides with very long half-lives may therefore be important. A recent epidemiological study suggests a higher chemical than radiological risk from [226]Ra [74], but it is also well known that uranium has a pronounced chemical toxicity that is more acute than its radiotoxicity [74, 75]. Thus, with secular equilibrium for [226]Ra with progenitor radionuclides, the large number of uranium atoms and covariation with the distribution of [226]Ra could help explain the statistical results. To be toxic, radionuclides must be bioavailable in aqueous solution and not adsorbed to grain and rock surfaces. The ratio between the amount of a radionuclide in aqueous solution compared to the amount of the radionuclide being adsorbed is known as the distribution coefficient ($K_d$), which is highly dependent on element, soil-type and soil parameters. For the study area, it has been reported that 77 to 94% of thorium and 38 to 69% of uranium present is irreversibly bound to soil, yielding arithmetic mean (AM) $K_d$s of 5600 (SD:4700) and 280 (SD: 320), respectively, suggesting low availability of thorium but high availability of uranium [44]. By comparison, in loam, sand and organic soils, uranium has some bioavailability with AM $K_d$ values of 2000–3000, while radium is less available in loam and sandy soils with AM $K_d$s of 8000–15000 but can be very available in organic soils with an AM $K_d$ of 200 [76]. It has been reported that OM has a high adsorption of radium [77], which would involve high ingestion rates of radium for soil organisms eating OM. In this study however, both radium isotopes were weakly and negatively correlated to OM. Adsorption and $K_d$ of radium is also negatively correlated to the concentration of other alkali metals like calcium due to competition for adsorption sites [76], and high soil calcium concentrations can involve radium $K_d$s as low as 5–100 [78]. Moreover, for uranium, both pH and carbonate are important, with maximal adsorption to soil at 5<pH<7 but with uranyl aqueous complexes forming at pH above 5 and carbonate complexes that increase bioavailability, especially at pH above 6 [76, 79]. This may be important, remembering the covariation of [226]Ra and uranium isotopes (given secular equilibrium) and the seeming overlap between [226]Ra and pH in explaining variation in litter loss. Moreover, as could be expected with the redrock in the study area, which consists of hematite-calcite-carbonatite, the cation exchange capacity is very high [44]. With the observed pH up to 5.5 (AM 4.8), and probably high calcium and carbonate levels, adsorption and $K_d$ is probably low for both [226]Ra and the uranium isotopes in the study area, highlighting the importance of uranium isotopes. The importance and large impact of uranium on soil fauna and decomposition rates have been demonstrated elsewhere [40, 41, 80], but there is also a general suggestion for more research on how different soil parameters affect the availability of radium isotopes [76]. Even though the very long half-life and large number of atoms of [238]U hints at the

importance of uranium toxicity, future research in the study area should assess how different soil parameters here affect both uranium and radium bioavailability.

At what level radionuclide dose rates will involve a negative effect on the litter decomposition of soil mesofauna is not known. The two litterbag studies from Chernobyl [36, 37] gave opposite results but were different in both radiation exposure, spatial coverage, and other methodology like litter species, drying regime, litter bag mesh size, soil parameters and statistical analyses. The study reporting effects used a handheld dosimeter detecting gamma dose rates from 0.1 to 240 µSv h$^{-1}$ and verified relative differences with government measurements [36]. The Chernobyl study that found no or positive effects, reported ambient radiation (gamma) from 0.2 to 29 µGy h$^{-1}$ and estimated total dose rates ranging from 0.3 to 150 µGy h$^{-1}$ [37]. The estimated total dose rates from the naturally occurring radionuclides in the study at hand range from 1.4 to 32 µGy h$^{-1}$ but the high degree of spatial heterogeneity and soil samples beneath all of the coarse litterbags allowed us to assess covariation of litter decomposition with highly variable radionuclide and soil parameter levels, enabling us to detect effects. Our statistical models indicate a negative effect and an explanatory power of dose rate from the naturally occurring radionuclides on variation in litter decomposition but also hints at effects of radionuclide toxicity. At Chernobyl, the anthropogenic radionuclides, $^{137}$Cs and $^{90}$Sr were the dominant ones [37]. Strontium is not considered toxic [81], and stable caesium at environmental levels is neither toxic to animals, even though both stable and radiocaesium may be somewhat chemically toxic at higher concentrations due to its similarity to kalium [82, 83]. Radiocaesium is thus probably much less chemically toxic than uranium and radium. This might suggest that in Chernobyl it is mainly dose and not any toxicity of radionuclides that are affecting soil fauna.

Regardless, the study at hand has identified effects on decomposition rate as expected from common soil parameters but also a negative effect of the occurrence of naturally occurring radionuclides that explain variation in litter loss from litterbags. Decomposition of OM and litter is an important soil ecosystem function and parameter [6, 8–12], which seems to be affected by changes in soil fauna biodiversity [84, 85]. With the effects seen on an ecosystem level in this study, we encourage future studies with an ecosystem approach [18, 86]. It may be useful to assess different ecosystem parts separately [18], and we suggest to preferentially address effects in earthworms, enchytraeids, collembolas and mites separately. Moreover, in areas like the study area, where soil may be as old as the last ice age, adaptation to radiation may have occurred, and it may be feasible to do irradiation studies of soil fauna extracted both from here and from control sites (where adaptation cannot have occurred since the ice age).

## Supporting information

**S1 Graphical abstract.**
(JPG)

## Acknowledgments

We thank the Geological Survey of Norway (NGU) and the Norwegian Mapping Authority (Kartverket) for the use of map data downloaded from the web pages Geonorge (https://kartkatalog.geonorge.no). We also thank Justin Brown for help with use of ERICA tool and helpful discussions on dosimetry.

## Author Contributions

**Conceptualization:** Hallvard Haanes, Runhild Gjelsvik.

**Data curation:** Hallvard Haanes, Runhild Gjelsvik.

**Formal analysis:** Hallvard Haanes.

**Funding acquisition:** Hallvard Haanes.

**Investigation:** Hallvard Haanes, Runhild Gjelsvik.

**Methodology:** Hallvard Haanes, Runhild Gjelsvik.

**Project administration:** Hallvard Haanes.

**Resources:** Hallvard Haanes, Runhild Gjelsvik.

**Software:** Hallvard Haanes.

**Supervision:** Hallvard Haanes.

**Validation:** Hallvard Haanes.

**Visualization:** Hallvard Haanes.

**Writing – original draft:** Hallvard Haanes, Runhild Gjelsvik.

**Writing – review & editing:** Hallvard Haanes, Runhild Gjelsvik.

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
