## [Decision Letter · Decision Letter 0]

19 Jan 2021

PONE-D-20-38999

Reduced soil fauna decomposition in a high background radiation area

PLOS ONE

Dear Dr. Haanes,

Thank you for submitting your manuscript to PLOS ONE. After careful consideration, we feel that it has merit but does not fully meet PLOS ONE’s publication criteria as it currently stands. Therefore, we invite you to submit a revised version of the manuscript that addresses the points raised during the review process.

This a wonderful experiment that attempts to get at several different issues related to decomposition and radiation in a fundamentally important ecological process, decomposition. The authors have put a lot of effort into their design and analyses and these findings are thus well worth reporting. There were however a few issues that will need to be addressed before the paper will be ready for publication. It is not clear to me if all of these issues can be addressed but the authors should do their best!

From my perspective, there were a  few important aspects of the study design that were not reported. First, although the idea of testing for differences in heat treatment (and thus extent of protein denaturalization) is a worthy question, it was not clear that this was addressed sufficiently. Was extent of denaturization measured?

Did the authors compare water content of leaf material from both oven and air dried samples? Air dried samples are unlikely to get any drier than ambient (unless placed in containers with drierite desiccant) while oven dried samples can get quite dry! Could this perhaps account for some of the findings?

I would like to see data on % loss reported as well. It looked like some of the bags (large mesh only) lost 100% of their material. Is this correct? 

Related to this, what did the authors do to minimize the loss of small particle sizes through the larger mesh holes during transport? What was done to minimize the loss due to mechanical factors over the course of the year as opposed to true decomposition effects? Were invertebrates counted in both bag types to demonstrate that this was indeed the cause of the differences?

The authors mention Chernobyl as a context for their work but I don't believe any comparisons are generated between radiation levels in Norway vs Chernobyl. My guess is that the differences are rather large…. How might this be reflected in the findings? Is the range of radiation levels used in this experiment great enough to test for radiation effects given the constraints on sample size?

The two reviewers also have a few suggestions for improvement that should be addressed in your revision.

We look forward to receiving your revised manuscript.

Kind regards,

Tim A. Mousseau

Academic Editor

PLOS ONE

Journal Requirements:

2. In your Methods section, please provide additional information regarding the permits you obtained to collect samples for the present study. Please ensure you have included the full name of the authority that approved the field site access and, if no permits were required, a brief statement explaining why.

"This work was supported by the Research Council of Norway through its Centres of Excellence

465 funding scheme, project number 223268/F50. This was received thorough the Centre for

466 Environmental Radioactivity, CERAD, Centre of Excellence CoE. We are thankful. We also thank the

467 Geological Survey of Norway (NGU) for use of map data. Thanks to Justin Brown for help with use of

468 ERICA tool and helpful discussions on dosimetry."

"NO - The funders had no role in study design, data collection and analysis, decision to publish, or preparation of the manuscript."

5. We note that Figure 1 in your submission contains map images which may be copyrighted. All PLOS content is published under the Creative Commons Attribution License (CC BY 4.0), which means that the manuscript, images, and Supporting Information files will be freely available online, and any third party is permitted to access, download, copy, distribute, and use these materials in any way, even commercially, with proper attribution. For these reasons, we cannot publish previously copyrighted maps or satellite images created using proprietary data, such as Google software (Google Maps, Street View, and Earth). For more information, see our copyright guidelines: http://journals.plos.org/plosone/s/licenses-and-copyright.

You may seek permission from the original copyright holder of Figure 1 to publish the content specifically under the CC BY 4.0 license. 

If you are unable to obtain permission from the original copyright holder to publish these figures under the CC BY 4.0 license or if the copyright holder’s requirements are incompatible with the CC BY 4.0 license, please either i) remove the figure or ii) supply a replacement figure that complies with the CC BY 4.0 license. Please check copyright information on all replacement figures and update the figure caption with source information. If applicable, please specify in the figure caption text when a figure is similar but not identical to the original image and is therefore for illustrative purposes only.

Reviewers' comments:

Reviewer's Responses to Questions

**Comments to the Author**

1. Is the manuscript technically sound, and do the data support the conclusions?

Reviewer #1: Yes

Reviewer #2: No

2. Has the statistical analysis been performed appropriately and rigorously? 

Reviewer #1: Yes

Reviewer #2: No

3. Have the authors made all data underlying the findings in their manuscript fully available?

Reviewer #1: Yes

Reviewer #2: No

4. Is the manuscript presented in an intelligible fashion and written in standard English?

Reviewer #1: Yes

Reviewer #2: Yes

5. Review Comments to the Author

Reviewer #1: The manuscript "Reduced soil fauna decomposition in a high background radiation area" by Hallvard Haanes and Runhild Gjelsvik, deals with the study of the radioactivity impact on soil fauna using the litter decomposition test. The topic is important due to needs to understand how the high radioactive contamination in areas of Chornobyl and Fukushima disasters as well as in areas of the natural enhanced radioactivity effects on ecosystem.

General comment

In general the manuscript is interesting, useful and good written; provides informative and good investigation. However, the manuscript should be slightly corrected. The sentences (Line 24) that "results on effects of anthropogenic radionuclides on soil fauna decomposition in Chernobyl are contradictory" and (L83-84) "litterbags have been used to both verify deleterious effects of anthropogenic radioactive pollution [33], as well as contradict them [34]" should be updated and clarified. Actually, if compare the statistics and range of study (by radioactivity level, spatial coverage) of two experiments in [33] and [34], rather results of [34] is doubtful. So I would propose authors to look at these papers more deep and update some details in these sentences.

The Chornobyl disaster radioactive contamination extended on Ukraine, Belarus and Russia areas. Therefore need to be more precise indicating the study area (see L85 "from Russian spill sites").

Minor comments

in Editorial Page 1

Abstract: (60 0 C) should be 60C

Order of Authors: ‘Haanes Gjelsvik’ should be Runhild Gjelsvik (see Manuscript: P1, L3)

in the manuscript

P3, L54 – Soil fauna affect should be Soil fauna affects

L70, 119, 122 – N:C or C:N – ??

P4, L76 – It has been show should be It has been shown

P4, L85 – Russian (??) – in Abstract [35]: several Soviet era nuclear disasters and the resulting severe radioactive pollution…

… over about 25,000 km2 of the territory of the former USSR alone. (see in General comment above)

P5, L100 – (580 mill y) – in SI, (580 Ma) (i.e. Mega annum) is preferable.

P5, L124 – September 28 – year have to be mentioned?

P7, L147 – The pattern of litterbags across localities was situated

P7, L150–151 but elsewise placements within each quartet were random

P7, L156 – October 17, – year?

P8, L187 – using 5 hours to reach (what??) and 12 hours at 550C – have to be corrected

P9, L191 – Cs137 – should be 137(Superscript)Cs

P9, L206–207 – while Enchytraeids range from 0.5 to 1.3 mm in width and 1 to 40 mm in length and earthworms range from 2 to 20 mm in with and 12 to 80 mm […].

should be: ‘in width and 12 to 80 mm in length […]’.

P9, L208 – ‘the concentration (CR) ratio equals’ should be ‘ the concentration ratio (CR) equals’

P10, L225 – ‘However, the for the tube-shape’

P11, L241 – log-transformed should be log(Subscript)10 transformed (subscript as in L257–260) or log(Subscript)10-transformed

P11, L247–250:

To assess spatial autocorrelation, correlations were made between a) an array of all possible pairwise physical distances all soil samples (n=1225) and b) each of arrays of the corresponding pairwise absolute differences and each of these soil sample pairs for each of the soil parameters. (check this sentence and clarify)

P16, L357–358 – Presence of macrofaunal like earthworms has a substantial effect on…

P18, L403 – (5.6) and L404 – 4.5e9. – units?

P18, L406 – A recent epidemiological study suggests…

More comments are included in the pdf text of the manuscript.

My opinion - the manuscript can be published after minor revision and corrections mentioned above.

Reviewer #2: The Authors set up a factorial experiment to understand the effect of mesh size and drying condition in determining decomposition in litter bags along a gradient of naturally occurring radionuclides. The starting point are: the lack of knowledge about the effects of such naturally-occurring radionuclides, and the existence of a controversy in Chernobyl, where different studies have reported decreased litter decomposition in sites of high radioactive contamination, or a lack of such effects. The authors conducted a good amount of work in their experiment, as well as in estimating the transfer of different radionuclides, and their contribution to radiation dose to different soil organisms.

However, the paper has fundamental shortcomings in all its major sections. The Introduction does not lay out a clear argument for the study, for example by explaining expected mechanisms for the reduction of decomposition following radiation exposure of the soil fauna. The Authors state in their abstract that they hypothesize naturally occurring radionuclides to affect soil fauna, and therefore decomposition. Yet they do not go into the precise mechanism –or alternative mechanisms – that might mediate such effect. They also routinely conflate radiation effects on abundance and diversity, and do not introduce the ecological roles of the different groups and their relevance to decomposition.

Surprisingly, the introduction does not even explain the specific results of previous studies (in Chernobyl and or elsewhere). Could variation among the results of previous studies be due to their methods? With the present paper being mostly methodological – which is fine –, I was surprised that the authors did not look more into the methods used by previous studies. The same is true for the effect of drying conditions, the other factor in their full factorial experiment.

In a similar fashion, the discussion also does not leverage estimates about radiation exposure toward shedding light on the effects of radiation on litter decomposition, in spite of this being the reported objective of the study.

The analytical approach is also problematic. For example, I'm struggling to understand what it means to be comparing different statistical models that include estimates of radiation dose to different organisms as predictors. In other general, the authors are using model comparison wrong. The comparisons should be based on a priori alternative hypotheses about how to explain variation in decomposition among litter bags. For example, by comparing a model that only includes physicochemical variables (i.e. organic matter, pH,…), With one that includes these variables plus variables that describe the biological community, and or the radiation does that the different components of that community received.

Instead, the Authors did stepwise model simplification too. This stepwise removal of statistically nonsignificant factors is known to lead to statistical errors. This is because statistical tests assume a single step, and are inappropriate when sequential steps are taken. This is not a found statistical approach, irrespective of whether the Authors also look at AIC. (See Whittingham et al. 2006 Methods in Ecology and Evolution)

Re: the structure of the models, I have some other doubts. For example, it looks like the authors are analyzing the two different types of litter bags (as defined by their mesh size) separately (L 333 and following). I thought that one of the main points was to compare the two types of litter bags, and was expecting this to be a factor in these linear models. I also think that the linear models should include site and/or location within sight as random factors.

This brings me to another crucial problem with the analyses described by the authors. From their description of how they compared decomposition rate in the different litter bags, it looks like they compare each litter bag with each other one, therefore greatly inflating the number of their comparisons.

Irrespective of the particular analyses chosen, the Authors do not clearly disentangle their results, or leverage them to advance understanding of the effect of . For example discriminating what explain the relevance of their results, and their

In addition, there is a certain lack of care throughout the paper, with several sentences that are broken (e.g., L 176) or riddled with typos

Some more specific comments follow.

Results

L 307 and following: All this variation in dosimetry, both within and among the different modeled organisms, is not analyzed or explained (it is also a missing component of the Discussion). It remains unclear whether such variation is due to uncertainty in the parameters, variation among different bags, or what.

Discussion

L 411: I suggested the authors actually lead with results in this paragraph. It's quite hard to follow the initial several sentences without a stated clear relationship with any result (which only appears on line 421). This paragraph will really benefit from a restructuring. Right now, it mentions several previous results, yet it never quite clarifies what the results of this paper add.

This is also true – as for the Introduction – with regards to the relationship with previous results. The Authors mention again the existence of a controversy in Chernobyl, yet the present results are not put into that context, nor the Authors discuss how their results help advance the status of knowledge there.

L 439: That the results indicate “an explanatory power of dose rate” is a vague way to put it. This is a common theme throughout. Results are often presented in this generic way, rather than explicitly indicating what the direction of the relationship is. (see for example: L 451: the study “identified effects”

L 445: ‘well below up to four times as high’ is a very confusing sentence.

L 447: ‘Radioceasium’ should be ‘radiocaesium’

L 449: I don’t understand this sentence. It is dose rates that are affecting soil fauna as opposed to what? Is this something about the dose rate as opposed to total overall dose?

L 455: I understand that the Authors did not measure biodiversity directly. Yet the relationship between biodiversity and decomposition is crucial to interpreting their results, as the Authors themselves hypothesize in the abstract and introduction, and acknowledge earlier in the discussion (e.g., L 443, when they talk about toxicity effects). So this should be expanded upon in the discussion. Right now, there is very little clarity as to what the results on litter decomposition indicate, as the Authors (1) do not sufficiently link them to the toxicity results (i.e., which taxa got the highest dose, and what that means for decomposition; which taxa were excluded by the mesh size, and what that means for decomposition; etc.).

Minor points

L 50: Agriculture is not an ecosystem function

L 80: I never heard anyone referring to Chernobyl as "a Russian spill site”

L 89: “does” should be “dose”

L 90: what kind of effects? This seems fairly important given the expected role of soil fauna in mediating litter decomposition. Yet, this is only vaguely referred to here.

L 170: Xxx?

L 179: define secular equilibrium

L 199: Ksi and chi are not defined anywhere

L 213—226: this entire paragraph would better be reported in a table

6. PLOS authors have the option to publish the peer review history of their article (what does this mean?). If published, this will include your full peer review and any attached files.

Reviewer #1: No

Reviewer #2: No

---

## [Author Response · Author response to Decision Letter 0]

12 Feb 2021

Dear editor and reviewers

We have now responded and accommodated to all comments. These are described in the rebuttal letter (separate file). We hope you are satisfied with our revision, it comes in two versions, one with track changes and one clean one.

---

## [Editor Report · Decision Letter 1]

15 Feb 2021

Reduced soil fauna decomposition in a high background radiation area

PONE-D-20-38999R1

Dear Dr. Haanes,

We’re pleased to inform you that your manuscript has been judged scientifically suitable for publication and will be formally accepted for publication once it meets all outstanding technical requirements. You have done a great job of responding to the reviews and I think you have addressed most of the concerns as best one could. So, congratulations.

Kind regards,

Tim A. Mousseau

Academic Editor

PLOS ONE

---

## [Editor Report · Acceptance letter]

19 Feb 2021

PONE-D-20-38999R1 

Reduced soil fauna decomposition in a high background radiation area 

Dear Dr. Haanes:

I'm pleased to inform you that your manuscript has been deemed suitable for publication in PLOS ONE. Congratulations! Your manuscript is now with our production department. 

Kind regards, 

on behalf of

Dr. Tim A. Mousseau 

Academic Editor

PLOS ONE